# Association of the Western Ontario and McMaster Universities Osteoarthritis Index (WOMAC) with Muscle Strength in Community-Dwelling Elderly with Knee Osteoarthritis

**DOI:** 10.3390/ijerph17072260

**Published:** 2020-03-27

**Authors:** Mi-Ji Kim, Byeong-Hun Kang, Soo-Hyun Park, Bokyoung Kim, Gyeong-Ye Lee, Young-Mi Seo, Ki-Soo Park, Jun-Il Yoo

**Affiliations:** 1Department of Preventive Medicine, College of Medicine and Institute of Health Science, Gyeongsang National University, Jinju 52725, Korea; mijikim@gnu.ac.kr (M.-J.K.); furim@gmail.com (S.-H.P.); parkks@gnu.ac.kr (B.K.); 2Center for Farmer’s Safety and Health, Gyeongsang National University Hospital, Jinju 52725, Korea; punggae@naver.com (G.-Y.L.); sechki486v@naver.com (Y.-M.S.); 3Department of Orthopaedic Surgery, Gyeongsang National University Hospital, Jinju 52725, Korea; furim@hanmail.net

**Keywords:** muscle strength, WOMAC, osteoarthritis

## Abstract

Purpose: The purpose of this study was to evaluate the correlation between muscle strength and knee symptoms (pain, stiffness, and functional limitation) regardless of the presence of radiologic knee osteoarthritis (RKOA) in community-dwelling elderly. Patients and methods: This cross-sectional study used data from the Namgaram-2 cohort. The Namgaram-2 cohort consisted of participants living in three rural communities. Such participants were included for studies on activity limitation due to age-related musculoskeletal disorders including knee osteoarthritis, osteoporosis, and sarcopenia. The Western Ontario and McMaster Universities Osteoarthritis Index (WOMAC), a health assessment tool for patients with arthritis in lower extremities, was used to assess health-related quality of life (HRQOL). Muscle strengths were measured by knee strength (by using the isokinetic dynamometer) and hand grip strength. Results: The WOMAC pain of Kallgren–Lawrence (K/L) grade < 2 was correlated with age, grip strength, nutrition status, and knee extension 180 peak torque. The WOMAC pain of K/L grade ≥ 2 was correlated with age, nutrition status, and knee extension 60 peak torque. The WOMAC stiffness of K/L grade < 2 was correlated with having a spouse, nutrition status, and knee extension 60 peak torque. The WOMAC stiffness of K/L grade ≥ 2 was correlated with knee extension 60 peak torque. The WOMAC function of K/L grade < 2 was correlated with age, grip strength, osteoporosis, nutrition status, and knee extension 180 peak torque. The WOMAC function of K/L grade ≥ 2 was correlated with age, nutrition status, and knee extension 60 peak torque. Conclusion: Muscle strength as measured by grip strength and knee extension was statistically significantly correlated with the WOMAC scores in patients with knee symptoms regardless of whether radiologic signs of knee osteoarthritis were observed.

## 1. Introduction

As aging continues, the most significant health behavior change in the elderly is physical impairment [1]. Proper limb function is important for physical activities [2,3]. One of the main causes of activity limitation in the elderly is knee arthritis [4]. However, it is still difficult to prevent or control the exacerbation of knee arthritis.

In addition, some patients complain of knee pain or dysfunction despite the absence of radiographic signs of knee arthritis. Still others experience no pain or dysfunction despite radiographic signs of knee osteoarthritis. Knee osteoarthritis is therefore not completely indicated by the presence or absence of radiographic signs.

Although there are many factors that affect symptomatic knee osteoarthritis, muscle weakness is associated with symptom complaints [5]. Recent weakness of the quadriceps muscle is an important cause of complaints [6], and knee extensor (KE) muscle strength is a modifiable factor of knee arthritis. Low KE strength has been associated with symptoms in patients with knee osteoarthritis [7]. Recent systematic reviews also suggest that low KE strength is associated with the development of radiological knee osteoarthritis (RKOA) [8]. In general, community epidemiological surveys recommend grip strength testing as an easy way to assess strength. In addition, research has demonstrated that this measured muscle strength is associated with pain in knee osteoarthritis [9].

Therefore, the purpose of this study was to evaluate the correlation between muscle strength and knee symptoms regardless of the presence of RKOA.

## 2. Subjects and Methods

### 2.1. Participants

This cross-sectional study used data from the Namgaram-2 cohort. The Namgaram-2 cohort consisted of a group of participants living in three rural communities. These participants were enrolled for studies on activity limitation due to work-related musculoskeletal disorders including knee osteoarthritis, osteoporosis, and sarcopenia. The subjects were residents aged 60 years or older who agreed to participate in this cohort from March 2016 to December 2018. A total 1010 people enrolled in the Namgaram-2 cohort answered questionnaires and received physical exams, blood tests, and radiographs. Participants were excluded if suffering from cognitive disorder, cardiovascular disease, or malignancy. This study was conducted based on data collected from the examinations. Participants included 890 individuals; 120 were excluded for either refusing to participate or for absence of upper and lower muscle strength assessments. All participants provided written informed consent. This study was approved by the Institutional Review Board of Gyeongsang National University (approval number: GIRB-A16-Y-0012).

## 3. Materials

The face-to-face survey was conducted by nurses who were aware of the purpose of this research and qualified in data collection procedures. Questionnaire completion required approximately 30 min. The survey included information on sociodemographic variables such as sex, age, presence or absence of a spouse, and smoking and alcohol consumption status. Nutritional condition was determined by the evaluation of the Mini Nutritional Assessment Short Form (MNA-SF). All participants were interviewed by educated investigators using the MNA-SF, which includes six items (range of scores: 0–14). According to the MNA-SF manual, the MNA-SF score is classified into two groups: well nourished (≥12 points) and risk of being malnourished (≤11 points). Blood tests included hemoglobin, uric acid, total cholesterol, γ-GTP (gamma-glutamyltransferase), CRP (C-reactive protein), and 25-OH vitamin D, all of which are related to nutritional status and RKOA.

### 3.1. Knee Pain and Function

The low extremity health assessment tool uses the Western Ontario and McMaster Universities Osteoarthritis Index (WOMAC) to assess pain, stiffness, and function of the lower extremity [10]. The WOMAC is a self-reported, lower extremity specific questionnaire and contains 24 questions: 17 on physical function, 5 on pain, and 2 on stiffness. Each query has five answer choices varying from 0 (no, without difficulty or no symptom) to 4 (unable to engage in activities or extreme symptoms). Subtotal scores for pain, stiffness, and function range from 0 to 20, 0 to 8, and 0 to 68, respectively. Total WOMAC scores were defined as the unweighted sums of all 24 items and ranged from 0 to 96.

### 3.2. Radiologic Knee Osteoarthritis (RKOA)

Radiographic assessment of both knees was performed, and RKOA was defined as level two or higher for at least one joint on the Kellgren–Lawrence (K/L) classification system. Radiographic images were interpreted by two radiologists, each with more than 20 years’ experience in musculoskeletal evaluation at a teaching hospital. When their opinions conflicted, discussion of their opinions was conducted until an agreement was reached. K/L grades were divided into 2 points or less and 2 or more points [11]. All images were examined with bilateral knee plain radiographs (bilateral anteroposterior and lateral; 30° of flexion, in a weight-bearing anteroposterior position) using a SD 3000 Synchro Stand.

### 3.3. Measurement of Body Composition

Dual energy X-ray absorptiometry (DEXA; Discovery W, Hologic, Waltham, MA, USA) was used to measure limb skeletal muscle index (SMI), which is obtained by dividing the appendicular skeletal mass (ASM) by the subject’s squared height (SMI = ASM/Ht²). For men, low muscle mass is defined as SMI < 7.0 kg/m²; for women, low muscle mass is < 5.4 kg/m² [12].

Bone density in the lumbar area was measured. A T-score of −2.5 or less was defined as osteoporosis. The coefficient of variation of the DEXA machine in our hospital was 2.2% (least significant change = 5.3%) for the lumbar spine measurements.

### 3.4. Measurement for the Hand Grip Strength (HGS) and Knee Strength

Since the measurement of HGS is one of the most commonly used methods for the diagnosis of sarcopenia, we defined the presence of sarcopenia based on HGS values [13]. HGS was measured using a digital hand dynamometer (Digital grip strength dynamometer, T.K.K 5401, Japan). The measurement of HGS was performed in a standing position with the forearm away from the body at the thigh level. Participants were asked to apply maximum HGS three times with both left and right hands. At least 30 s of resting interval was allowed between each measurement. HGS was defined as maximally measured grip strength of the dominant hand [13]. Low HGS was defined according to the Asian Working Group for Sarcopenia (AWGS) criteria for low muscle mass strength (hand grip strength below 18 kg in women and below 26 kg in men) [12].

Knee strength was measured by using the isokinetic dynamometer (Biodex Multi-Joint System 4; Biodex Medical Systems, Inc., Shirley, NY, USA). The examiner placed a knee stabilizer pad on the subject’s ankle and fixed the chest, abdomen, and femur with a band so that no external force was applied to the pelvis and femoral muscle movements. The knee extension 180 peak torque and knee extension 60 peak torque were measured at the knee joint. In the measurement of knee strength, the gravity effect torque was first calculated to exclude the influence of leg weight on the muscle strength. The measurement of knee strength was divided into dominant side and nondominant side; the loading speed was 60°/sec for 4 repetitions [14].

### 3.5. Statistical Analysis

Continuous variables were expressed as means and standard deviations, and nominal variables were expressed as numbers and percentages. The Pearson’s correlation test was used to evaluate the relationship between presence of HRQOL (the WOMAC scores) and muscle strength. Also, multiple regression analysis was used to assess the association between the WOMAC scores and muscle strength. There was multicollinearity between the knee extension 60 peak torque and the knee extension 180 peak torque (variance inflation factor (VIF) = 0.806–0.817). Therefore, the stepwise method was used for variable selection in the multiple regression analyses, so that only the significant variables were selected first. Included variables were sex (male, female: reference), age (years), spouse (presence, absence: reference), smoking status (current smoker, nonsmoker: reference), alcohol consumption status (drinking alcohol, not drinking alcohol: reference), SMI (low, robust: reference), grip strength (low, robust: reference; we used grip strength as a continuous variable in the correlation analysis), osteoporosis (osteoporosis, robust: reference), and nutrition status (risky of being malnourished, well nourished: reference). Laboratory examination variables were also included: hemoglobin (g/dL), uric acid (mg/dL), total cholesterol (mg/dL), γ-GTP (IU/L), and 25-OH vitamin D (ng/mL). We assessed knee extension strength variables, measured by knee extension 60 peak torque (Nm/kg) and knee extension 180 peak torque (Nm/kg). The Cohen’s kappa correlation coefficient for agreement between the evaluators was 0.839 for plane radiographs, suggesting excellent level of agreement. All statistical analyses were carried out using the SPSS version 23.0 software (SPSS Inc., Chicago, IL, USA), and *p*-values < 0.05 were defined as statistically significant.

## 4. Results

### 4.1. Demographics and Clinical Characteristics of the Study Population

The mean age of the subjects was 67.5 ± 7.5 years old for the K/L grade < 2 group and 70.9 ± 6.7 years old for the K/L grade ≥ 2 group. Most people were female in the K/L ≥ 2 group (85.3%), while 58.8% of the population were female in the K/L < 2 group. In the K/L ≥ 2 group, 42.5% of people did not have their spouse, while 27.9% did not in the K/L < 2 group. The total WOMAC scores were 15.9 ± 17.2 for the K/L < group and 27.6 ± 19.3 for the K/L ≥ group (Table 1).

### 4.2. Correlation of WOMAC and Muscle Strength Measured by Grip Strength and Knee Extensor Strength

In the group with K/L < 2, WOMAC pain, WOMAC stiffness, WOMAC function, and WOMAC total were statistically significantly correlated negatively with grip strength, knee extension 60 peak torque, and knee extension 180 peak torque (*p* < 0.001).

In the group with K/L ≥ 2, WOMAC pain, WOMAC stiffness, WOMAC function, and WOMAC total were statistically significantly correlated negatively with grip strength, knee extension 60 peak torque, and knee extension 180 peak torque (*p* < 0.001) (Table 2).

### 4.3. Results of Stepwise Multiple Regression Analysis for the WOMAC Pain

The WOMAC pain of K/L < 2 was positively correlated with age (*p* = 0.006), osteoporosis (*p* = 0.012), risk of being malnourished (*p* = 0.002), and low grip strength (*p* < 0.001) whereas negatively correlated with knee extension 180 peak torque (*p* < 0.001). The WOMAC pain of K/L ≥ 2 was also positively correlated with age (*p* < 0.046) and risk of being malnourished (*p* < 0.001) whereas negatively correlated with knee extension 60 peak torque (*p* < 0.001) (Table 3).

### 4.4. Results of Stepwise Multiple Regression Analysis for the WOMAC Stiffness

The WOMAC stiffness of K/L < 2 was positively correlated with having a spouse (*p* = 0.047) and weak grip strength (*p* < 0.001) and negatively correlated with knee extension 60 peak torque (*p* < 0.001). The WOMAC stiffness of K/L ≥ 2 was positively correlated with risky nutrition status (*p* < 0.001) and negatively correlated with knee extension 60 peak torque (*p* < 0.001) (Table 4).

### 4.5. Results of Stepwise Multiple Regression Analysis for the WOMAC Function

The WOMAC function of K/L < 2 was positively correlated with age (*p* = 0.003), having a spouse (*p* = 0.005), having osteoporosis (*p* = 0.022), risky nutrition status (*p* = 0.002), and low grip strength (*p* < 0.001) while negatively correlated with knee extension 180 peak torque (*p* < 0.001). The WOMAC function of K/L ≥ 2 was positively correlated with age (*p* = 0.003) and risky nutrition status (*p* < 0.001) while negatively correlated with knee extension 60 peak torque (*p* < 0.001) (Table 5).

### 4.6. Results of stepwise multiple regression analysis for the WOMAC total

The WOMAC total of K/L < 2 was positively associated with age (*p* = 0.006), osteoporosis (*p* = 0.012), risk of malnutrition (*p* = 0.002), and low grip strength (*p* < 0.001) whereas negatively associated with knee extension 180 peak torque (*p* < 0.001). The WOMAC total of K/L ≥ 2 was positively related with age (*p* = 0.006) and risk of malnutrition (*p* < 0.001) whereas negatively related with knee extension 60 peak torque (*p* < 0.001) (Table 6).

## 5. Discussion

The principle findings of this study were that muscle strength (grip and knee extensor strength) was statistically associated with WOMAC in the group with no RKOA and that knee extension was statistically associated with WOMAC in patients with RKOA.

The most widely used condition-specific instrument for the assessment of OA of the lower extremities is the WOMAC scores; these scores have been determined to be valid and reliable [15]. In our study, those with K/L grade 2 or higher were significantly more likely to have higher total WOMAC scores (27.6 ± 19.3) than those with K/L grade 1 or lower (15.9 ± 17.2). In addition, 84.8% of the respondents answered to having symptoms even in the K/L grades below 2. Although not RKOA, many groups complain of symptoms. In each detailed area of WOMAC, 80.7% of the groups had a slight impairment of function. When assessing leg-related function, knee arthritis also requires muscle evaluation.

Especially, the knee extension 60 peak torque and the knee extension 180 peak torque showed negative association with the WOMAC function of K/L < 2. Whether knee extensor muscle weakness is a risk factor for knee osteoarthritis is important and needs to be confirmed as muscle strength is a potential modifiable risk factor. Muscle strength is related to knee pain and disability rather than RKOA. In the definition of sarcopenia, low muscle function is usually assessed by slow walking speed or low handgrip strength. Both features are associated with increased morbidity, falls, and mortality in older people [16]. The weakening of grip strength to measure sarcopenia was statically significantly associated with a high WOMAC score. As our study results demonstrate, lower muscle strength was associated with WOMAC score.

Recent studies have reported that knee pain or functional limitations are often present despite the absence of RKOA [17,18]. O’Reilly et al. [19] reported that knee OA pain was closely related to weakness of the quadriceps muscle. Messier [6] also reported that knee pain correlated with quadriceps muscle weakness. In addition, Ruhdorfer et al., reported that reduction in thigh muscle strength in knee OA was related to pain but not to radiographic (K/L grade) status [17]. A recent meta-analysis also reported that weakness of the knee extensor muscle was associated with an increased risk of developing knee osteoarthritis in both men and women [8].

The quadricep muscle absorbs the impact on the joints and stabilizes the loaded legs. Weakness of the quadriceps muscles caused insufficient impact on the lower extremities; this created excessive stress on the lower extremities causing knee pain [20]. Segal and Glass [21] concluded that greater quadricep muscle strength protected against symptomatic but not radiographic knee osteoarthritis in men and women. Berry and Cicuttini [22,23] have shown that skeletal muscle mass protects against OA onset and observed a positive association between skeletal muscle mass and joint space width [24]. However, the relationship between skeletal muscle strength and symptoms remains unclear.

In this study, in addition to muscle strength, nutritional status was also significantly associated with WOMAC. Recently, sarcopenia is considered an important prognostic factor for osteoarthritis, [25,26] especially knee osteoarthritis. Considering that one of the risk factors for sarcopenia is malnutrition, sarcopenia due to malnutrition continues and eventually affects the symptoms of knee osteoarthritis. In particular, the elderly had more functional improvement effect when exercising together with nutrition, so proper nutrition and muscle strengthen exercise are needed together [27].

In addition, it was reported that WOMAC stiffness had a negative association with the spouse. If the study subjects had a spouse, in the end, they may have had a bad effect on knee osteoarthritis because they performed a lot of agricultural work than subjects without a spouse. However, follow-up studies will have to be conducted to prove these results.

There are several limitations to this study. First, the study was based on cross-sectional data. This restricted our ability to determine a causal relationship between patients’ perceptions of regional musculoskeletal status and general health status. Exploring the causal relationship would require considering complex biological and psychologic factors. Second, our cohort was limited to a rural area population. Therefore, the data may not be representative of other elderly populations. Third, testing the BMD was performed at a single site: lumbar spine. Therefore, the prevalence of osteoporosis may have been underestimated.

In conclusion, grip and knee extension muscle strength were statistically correlated with WOMAC in patients with no RKOA. Therefore, elderly people in the community will need muscle strengthening exercises and proper nutrition will also be important.

## Figures and Tables

**Table 1 ijerph-17-02260-t001:** Demographics and clinical characteristics of the study population (N = 891).

Characteristics	K/L < 2 (N = 631)	K/L ≥ 2 (N = 259)
Sex, n (%)		
Male	260 (41.2)	38 (14.7)
Female	371 (58.8)	221 (85.3)
Age, years	67.5 ± 7.5	70.9 ± 6.7
Spouse, n (%)		
Presence	455 (72.1)	149 (57.5)
Absence	176 (27.9)	110 (42.5)
Smoking status, n (%)		
Current smoker	55 (8.7)	12 (4.6)
Nonsmoker	576 (91.3)	247 (95.4)
Alcohol consumption status, n (%)		
Drinking alcohol	39 (6.2)	7 (2.7)
Not drinking alcohol	592 (93.8)	252 (97.3)
SMI, n (%)		
Low	478 (75.8)	207 (79.9)
Robust	153 (24.3)	52 (20.1)
Grip strength, n (%)		
Low	143 (22.7)	93 (35.9)
Robust	488 (77.3)	166 (64.1)
Osteoporosis, n (%)		
Osteoporosis	93 (14.7)	41 (15.8)
Robust	538 (85.3)	218 (84.2)
Nutrition status, n (%)		
Risky of being malnourished	91 (14.4)	46 (17.8)
Well nourished	540 (85.6)	213 (82.2)
Laboratory examination		
Hemoglobin, g/dL	13.6 ± 1.5	13.1 ± 1.3
Uric acid, mg/dL	4.8 ± 1.4	4.7 ± 1.2
Total cholesterol, mg/dL	185.4 ± 37.1	187.9 ± 38.5
γ-GTP, IU/L	35.3 ± 63.7	28.4 ± 32.1
CRP, mg/L	1.7 ± 4.5	1.5 ± 3.8
25-OH vitamin D, ng/mL	29.0 ± 10.9	27.2 ± 10.4
Knee extension strength		
Knee extension 60 peak torque, Nm/kg	69.0 ± 30.7	50.3 ± 23.9
Knee extension 180 peak torque, Nm/kg	43.3 ± 19.4	31.8 ± 13.9
WOMAC		
WOMAC pain	3.0 ± 3.6	5.5 ± 4.4
WOMAC pain ≥ 0, n (%)	415 (65.8)	234 (90.4)
WOMAC stiffness	1.3 ± 1.6	2.1 ± 2.0
WOMAC stiffness ≥ 0, n (%)	330 (52.3)	184 (71.0)
WOMAC function	11.6 ± 12.8	20.0 ± 14.0
WOMAC function ≥ 0, n (%)	509 (80.7)	247 (95.4)
WOMAC total	15.9 ± 17.2	27.6 ± 19.3
WOMAC total ≥ 0, n (%)	535 (84.8)	248 (95.8)

K/L, Kellgren-Lawrence grade; SMI, skeletal muscle index; γ-GTP, gamma-glutamyltransferase; CRP, C-reactive protein; 25-OH vitamin D, 25-hydroxy vitamin D; WOMAC, Western Ontario and McMaster Universities Osteoarthritis Index. Values are presented as means ± standard deviations or numbers (percentages).

**Table 2 ijerph-17-02260-t002:** Correlation analysis between WOMAC and muscle strength measured by grip strength and knee extensor strength.

	WOMAC Pain		WOMAC Stiffness		WOMAC Function		WOMAC Total	
	K/L < 2	K/L ≥ 2	K/L < 2	K/L ≥ 2	K/L < 2	K/L ≥ 2	K/L < 2	K/L ≥ 2
Grip strength	−0.422 (<0.001)	−0.324 (<0.001)	−0.310 (<0.001)	−0.188 (0.004)	−0.489 (<0.001)	−0.358 (<0.001)	−0.482 (<0.001)	−0.352 (<0.001)
Knee extension 60 peak torque	−0.419 (<0.001)	−0.417 (<0.001)	−0.293 (<0.001)	−0.245 (<0.001)	−0.445 (<0.001)	−0.376 (<0.001)	−0.447 (<0.001)	−0.392 (<0.001)
Knee extension 180 peak torque	−0.426 (<0.001)	−0.404 (<0.001)	−0.293 (<0.001)	−0.241 (<0.001)	−0.453 (<0.001)	−0.366 (<0.001)	−0.454 (<0.001)	−0.381 (<0.001)

K/L, Kellgren–Lawrence grade; WOMAC, Western Ontario and McMaster Universities Osteoarthritis Index. Values are presented as Pearson’s correlation coefficients (p-values). Grip strength was used as a continuous variable in this correlation analysis.

**Table 3 ijerph-17-02260-t003:** Results of stepwise multiple regression analysis for the WOMAC pain.

WOMAC Pain	K/L < 2			K/L ≥ 2		
	95% CI for Estimate		*p*	95% CI for Estimate		*p*
	Lower	Upper		Lower	Upper	
Age (years)	0.073	0.422	0.006	0.002	0.153	0.046
Osteoporosis (osteoporosis/robust)	0.928	7.581	0.012			
Nutrition status (risky/well nourished)	2.013	8.671	0.002	1.663	4.155	<0.001
Grip strength (low/robust)	4.374	10.646	<0.001			
Knee extension 180 peak torque (Nm/kg)	−0.344	−0.209	<0.001			
Knee extension 60 peak torque (Nm/kg)				-0.082	-0.039	<0.001

K/L, Kellgren–Lawrence grade; WOMAC, Western Ontario and McMaster Universities Osteoarthritis Index; 95% CI, 95% confidence interval. *p*-values were determined by stepwise multiple regression analysis. Age, knee extension 180 peak torque, and knee extension 60 peak torque were continuous variables. Osteoporosis (reference group: no), nutrition status (reference group: well nourished), and grip strength (reference group: robust) were categorical variables.

**Table 4 ijerph-17-02260-t004:** Results of stepwise multiple regression analysis for the WOMAC stiffness.

WOMAC Stiffness	K/L < 2			K/L ≥ 2		
	95% CI for Estimate		*p*	95% CI for Estimate		*p*
	Lower	Upper		Lower	Upper	
Spouse (presence/absence)	0.004	0.578	0.047			
Nutrition status (risky/well nourished)				0.546	1.778	<0.001
Grip strength (low/robust)	0.257	0.894	<0.001			
Knee extension 60 peak torque (Nm/kg)	−0.015	−0.006	<0.001	−0.027	−0.007	<0.001

K/L, Kellgren–Lawrence grade; WOMAC, Western Ontario and McMaster Universities Osteoarthritis Index; 95% CI, 95% confidence interval. *p*-values were determined by stepwise multiple regression analysis. Knee extension 60 peak torque was a continuous variable. Spouse (reference group: no), nutrition status (reference group: well nourished), and grip strength (reference group: robust) were categorical variables.

**Table 5 ijerph-17-02260-t005:** Results of stepwise multiple regression analysis for the WOMAC function.

WOMAC Function	K/L < 2			K/L ≥ 2		
	95% CI for Estimate		*p*	95% CI for Estimate		*p*
	Lower	Upper		Lower	Upper	
Age (years)	0.066	0.327	0.003	0.133	0.622	0.003
Osteoporosis (osteoporosis/robust)	0.419	5.375	0.022			
Nutrition status (risky/well nourished)	1.394	6.376	0.002	4.534	12.584	<0.001
Grip strength (low/robust)	3.451	8.128	<0.001			
Knee extension 180 peak torque (Nm/kg)	−0.248	−0.147	<0.001			
Knee extension 60 peak torque (Nm/kg)				−0.226	−0.089	<0.001

K/L, Kellgren–Lawrence grade; WOMAC, Western Ontario and McMaster Universities Osteoarthritis Index; 95% CI, 95% confidence interval. *p*-values were determined by stepwise multiple regression analysis. Age, knee extension 180 peak torque, and knee extension 60 peak torque were continuous variables. Osteoporosis (reference group: no), nutrition status (reference group: well nourished), and grip strength (reference group: robust) were categorical variables.

**Table 6 ijerph-17-02260-t006:** Results of stepwise multiple regression analysis for the WOMAC total.

WOMAC Total	K/L < 2			K/L ≥ 2		
	95% CI for Estimate		*p*	95% CI for Estimate		*p*
	Lower	Upper		Lower	Upper	
Age (years)	0.073	0.422	0.006	0.135	0.807	0.006
Osteoporosis (osteoporosis/robust)	0.928	7.581	0.012			
Nutrition status (risky/well nourished)	2.013	8.671	0.002	7.077	18.126	<0.001
Grip strength (low/robust)	4.374	10.646	<0.001			
Knee extension 180 peak torque (Nm/kg)	−0.344	−0.209	<0.001			
Knee extension 60 peak torque (Nm/kg)				−0.327	−0.139	<0.001

K/L, Kellgren–Lawrence grade; WOMAC, Western Ontario and McMaster Universities Osteoarthritis Index; 95% CI, 95% confidence interval. *p*-values were determined by stepwise multiple regression analysis. Age, knee extension 180 peak torque, and knee extension 60 peak torque were continuous variables. Osteoporosis (reference group: no), nutrition status (reference group: well nourished), and grip strength (reference group: robust) were categorical variables.

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
