# Peer review of "Association of the Western Ontario and McMaster Universities Osteoarthritis Index (WOMAC) with Muscle Strength in Community-Dwelling Elderly with Knee Osteoarthritis"

_ijerph, 2020, doi:10.3390/ijerph17072260_

Round 1
Reviewer 1 Report
This is a good paper.
Let me ask one question. Why do the authors select cross sectional study?
Author Response
Thank you for you valuable comment.
Reviewer 2 Report
The manuscript is lacking of novelty as it is expected that patients with osteoarthritis will suffer from a higher WOMAC score. The authors may want to highlight the research gap in the relationship between muscle strength and osteoarthritis in this case. The language used in this manuscript requires major revisions to improve readability. Title: Do not abbreviate WOMAC in the title. Add '-' between community and dwelling Abstract: The authors used NAMGARAM-2 in the abstract and Namgaram-2 in the text. Which one is correct? Is it an abbreviation? Please use the full term for HRQOL Introduction: The statement "as aging continues... is physical activity" is not clear. Did the authors mean "The most significant health behaviour change in the elderly is physical impairment"? What did the author mean by "methods to prevent or control the progression or deterioration of knee arthritis are difficult"? Methods: did the authors test the kappa agreement between the two evaluators in evaluating the knee radiographs? Testing the BMD at the single site 'lumbar spine' may reduce the prevalence of osteoporosis. This should be listed as one of the limitations. Please also report the Cv values of the DXA machine used. Please add '-' between T and score. The authors indicated that a multicollinearity issue was noted in the regression, so how did they solve this issue? The sentence "...WOMAC total was statistically significant with grip strength..." was not meaningful. An adjustive or verb is missing in the sentence. Please report the direction of the association in the correlation and regression analysis. For example, XX was correlated positively with XX. Please note the real p-value in the correlation table. For the categorical predictor in the regression model, it is necessary for the authors to mention which was the reference group. Discussion: line 215: "the knee extension .... were statistically significant" again, this sentence is not meaningful, please revise.Author Response
Reviewer 2
The manuscript is lacking of novelty as it is expected that patients with osteoarthritis will suffer from a higher WOMAC score. The authors may want to highlight the research gap in the relationship between muscle strength and osteoarthritis in this case. The language used in this manuscript requires major revisions to improve readability.
Title: Do not abbreviate WOMAC in the title. Add '-' between community and dwelling
Response: We changed “Association of WOMAC with muscle strength in community dwelling elderly with Knee Osteoarthritis” to “Association of the Western Ontario and McMaster Universities Osteoarthritis Index (WOMAC) with Muscle Strength in Community-dwelling Elderly with Knee Osteoarthritis” in the Titles.
Abstract: The authors used NAMGARAM-2 in the abstract and Namgaram-2 in the text. Which one is correct? Is it an abbreviation? Please use the full term for HRQOL
Response: Namgaram is a local name and is a proper noun, not an abbreviation. Therefore, we changed “NAMGARAM-2” to “Namgaram-2” in the whole manuscript.
In addition, we changed “HRQOL” to “health related quality of life (HRQOL)” in the Abstract.
Introduction: The statement "as aging continues... is physical activity" is not clear. Did the authors mean "The most significant health behaviour change in the elderly is physical impairment"?
Response: As your advice, we changed “As aging continues, the most significant health behavior among elderly individuals is physical activity.” to “The most significant health behavior change in the elderly is physical impairment.” in the Introduction.
What did the author mean by "methods to prevent or control the progression or deterioration of knee arthritis are difficult"?
Response: We changed “methods to prevent or control the progression or deterioration of knee arthritis are difficult" to “it is still difficult to prevent or control the exacerbation of knee arthritis.” in the Introduction.
Methods: did the authors test the kappa agreement between the two evaluators in evaluating the knee radiographs?
Response: We described “The Cohen's kappa correlation coefficient for agreement between the evaluators was 0.839 for plane radiographs, suggesting excellent level of agreement.” in the Methods.
Testing the BMD at the single site 'lumbar spine' may reduce the prevalence of osteoporosis. This should be listed as one of the limitations.
Response: We described “Third, testing the BMD was performed at the single site, lumbar spine. Therefore, the prevalence of osteoporosis may have been underestimated.” in the limitation of the Discussion.
Please also report the Cv values of the DXA machine used.
Response: We described “The coefficient of variation of the DEXA machine in our hospital was to 2.2 % (least significant change = 5.3%) for the lumbar spine measurements.” in the Methods.
Please add '-' between T and score.
Response: We changed “T score” to “T-score” in the Methods.
The authors indicated that a multicollinearity issue was noted in the regression, so how did they solve this issue?
Response: We described “There was multicollinearity between the knee extension 60 peak torque and the knee extension 180 peak torque (variance inflation factor (VIF) = 0.806 - 0.817). Therefore, the stepwise method was used for variable selection in the multiple regression analyses, so that only the significant variables were selected first.” in the Methods.
The sentence "...WOMAC total was statistically significant with grip strength..." was not meaningful. An adjustive or verb is missing in the sentence.
Response: We have revised the sentences “…total WOMAC score were statistically significantly correlated with grip strength, …” in the Results section.
Please report the direction of the association in the correlation and regression analysis. For example, XX was correlated positively with XX.
Response: We have reported the direction of the association in the Results section (e.g., “In the group with K/L < 2, WOMAC pain, WOMAC stiffness, WOMAC function, and WOMAC total were statistically significantly correlated negatively with grip strength, knee extension 60 peak torque, and knee extension 180 peak torque (p < 0.001).”).
Please note the real p-value in the correlation table.
Response: We have noted the p-values in the correlation table (Table 2).
For the categorical predictor in the regression model, it is necessary for the authors to mention which was the reference group.
Response: We have revised the Method section (“Included variables were sex (male, female: reference), age (years), spouse (presence, absence: reference), smoking status (current smoker, non-smoker: reference), alcohol consumption status (drinking alcohol, not drinking alcohol: reference), SMI (low, robust: reference), grip strength (low, robust: reference; we used grip strength as a continuous variable in the correlation analysis), osteoporosis (osteoporosis, robust: reference), and nutrition status (risky of being malnourished, well nourished: reference).”) We have added variable details under the tables (Table 3, 4, 5, and 6) in the Results section (e.g., “Age, knee extension 180 peak torque, and knee extension 60 peak torque were continuous variables. Osteoporosis (reference group: no), nutrition status (reference group: well nourished), and grip strength (reference group: robust) were categorical variables.”).
Discussion: line 215: "the knee extension .... were statistically significant" again, this sentence is not meaningful, please revise.
Response: We changed the sentence “Especially, the knee extension 60 peak torque and the knee extension 180 peak torque showed negative association with the WOMAC function of K/L <2.” in the Discussion.

Reviewer 3 Report
This is a very good correlative study.
Some minor improvements recommended:
- The font size is not same throughout the entire manuscript.
- Could expand on RKOA, as to which radiographic technique was used.
- The discussion should elaborate more on the results than on the literature review. For example any reasoning on why the WOMAC stiffness 36 of K/L grade < 2 was correlated with not having a spouse
- Also include recommendations for these patients on their lifestyle, nutritional status based on the findings from this study.
Author Response
Reviewer 3
This is a very good correlative study.
Some minor improvements recommended:
- The font size is not same throughout the entire manuscript.
Response: We corrected the font and font size throughout the entire manuscript.
- Could expand on RKOA, as to which radiographic technique was used.
Response: We described “All images were examined with bilateral knee plain radiographs (bilateral anteroposterior and lateral; 30° of flexion, in a weight-bearing anteroposterior position) using a SD 3000 Synchro Stand.” in the Methods.
- The discussion should elaborate more on the results than on the literature review. For example any reasoning on why the WOMAC stiffness 36 of K/L grade < 2 was correlated with not having a spouse
Response: We have described “In this study, in addition to muscle strength, nutritional status was also significantly associated with WOMAC. Recently, sarcopenia is considered to be an important prognostic factor for osteoarthritis, especially knee osteoarthritis. Considering that one of the risk factors for sarcopenia is malnutrition, sarcopenia due to malnutrition continues and eventually affects the symptoms of knee osteoarthritis. In particular, the elderly had more functional improvement effect when exercising together with nutrition, so proper nutrition and muscle strengthen exercise are needed together.
In addition, it was reported that WOMAC stiffness had a negative association with the spouse. If the study subjects had a spouse, they were mostly women, and in the end, they may have had a bad effect on knee osteoarthritis because they performed a lot of agricultural work than women without a spouse. However, follow-up studies will have to be conducted to prove these results.” in the Discussion.
- Also include recommendations for these patients on their lifestyle, nutritional status based on the findings from this study.
Response: We have described “Therefore, elderly people in the community will need muscle strengthening exercises and proper nutrition will also be important.” in the Discussion.

Round 2
Reviewer 2 Report
Thank you for addressing my comments. Since the previous comments are answered adequately, I have no further comments for this manuscript.
Thank you.